# Rare Occurrence of Aristolochic Acid Mutational Signatures in Oro-Gastrointestinal Tract Cancers

**DOI:** 10.3390/cancers14030576

**Published:** 2022-01-24

**Authors:** Abner Herbert Lim, Jason Yongsheng Chan, Ming-Chin Yu, Tsung-Han Wu, Jing Han Hong, Cedric Chuan Young Ng, Zhen Jie Low, Wei Liu, Rajasegaran Vikneswari, Pin-Cheng Sung, Wen-Lang Fan, Bin Tean Teh, Sen-Yung Hsieh

**Affiliations:** 1Cheng Kin Ku Herbal Biodiversity & Medicine Program, SingHealth Duke-NUS Institute of Biodiversity Medicine, Singapore 169610, Singapore; abner.lim.m.s@nccs.com.sg (A.H.L.); jinghan.hong@duke-nus.edu.sg (J.H.H.); cedric.ng.c.y@nccs.com.sg (C.C.Y.N.); lowzhenjie@hotmail.com (Z.J.L.); liu.wei@nccs.com.sg (W.L.); 2Cancer Discovery Hub, National Cancer Centre Singapore, Singapore 169610, Singapore; jason.chan.y.s@nccs.com.sg (J.Y.C.); vikneswari.r@nccs.com.sg (R.V.); 3Laboratory of Cancer Epigenome, National Cancer Centre Singapore, Singapore 169610, Singapore; 4Division of Medical Oncology, National Cancer Centre Singapore, Singapore 169610, Singapore; 5Department of Surgery, Chang Gung Memorial Hospital, Linkou, Taoyuan 333, Taiwan; a75159@cgmh.org.tw (M.-C.Y.); domani@cgmh.org.tw (T.-H.W.); 6Department of Surgery, New Taipei Municipal Tucheng Hospital, New Taipei City 236, Taiwan; 7College of Medicine, Chang Gung University, Taoyuan 333, Taiwan; 8Cancer and Stem Cell Biology Program, Duke-NUS Medical School, Singapore 169857, Singapore; 9Department of Gastroenterology and Hepatology, Chang Gung Memorial Hospital, Linkou, Taoyuan 333, Taiwan; dannysung24@cgmh.org.tw; 10Genomic Medicine Core Laboratory, Chang Gung Memorial Hospital, Linkou, Taoyuan 333, Taiwan; wlfan@cgmh.org.tw; 11Oncology Academic Clinical Program, Duke-NUS Medical School, Singapore 169857, Singapore

**Keywords:** carcinogens, mutagenesis, next generation sequencing, genomics, aristolochic acid

## Abstract

**Simple Summary:**

Aristolochic acids (AAs) are a family of carcinogenic phytochemical compounds commonly found in plants of the Aristolochia and Asarum genera. Comprehensive genomic profiling of genitourinary and hepatobiliary cancers has highlighted the widespread prevalence of Aristolochic acid (AA) signatures in cancer patients across parts of Asia, particularly in Taiwan. The aim of our study was to determine in oro-gastrointestinal tract cancers (OGITCs), the prevalence, role and significance that AA plays as a driver of tumorigenesis as AA-containing products are commonly administered orally. This suggests a possible etiological relationship between OGITCs. However, in this study, the rarity of AA mutational signatures in OGITCs suggests that AA is unlikely to drive carcinogenesis in OGITCs through direct exposure. Our study is valuable because it shows that AA exposure is not an equal driver of tumorigenesis in different organs and represents an important piece of information in the field.

**Abstract:**

Background: Aristolochic acids (AAs) are potent mutagens commonly found in herbal plant-based remedies widely used throughout Asian countries. Patients and Methods: To understand whether AA is involved in the tumorigenesis of the oro-gastrointestinal tract, we used whole-exome sequencing to profile 54 cases of four distinct types of oro-gastrointestinal tract cancer (OGITC) from Taiwan. Results: A diverse landscape of mutational signatures including those from DNA mismatch repair and reactive oxygen species was observed. APOBEC mutational signatures were observed in 60% of oral squamous cell carcinomas. Only one sample harbored AA mutational signatures, contradictory to prior reports of cancers from Taiwan. The metabolism of AA in the liver and urinary tract, transient exposure time, and high cell turnover rates at OGITC sites may explain our findings. Conclusion: AA signatures in OGITCs are rare and unlikely to be a major contributing factor in oro-gastrointestinal tract tumorigenesis.

## 1. Introduction

Cancers involving the oro-gastrointestinal tract (OGITCs) account for amongst the highest rates of leading cancers worldwide in both sexes. Globally, the prevalence of OGITCs is estimated to be at 4.8 million, with 3.4 million related deaths based on statistics gathered in 2018, with esophageal and gastric cancers leading in Asia [1]. Factors such as ageing [2], exposure to carcinogens (e.g., smoking [3,4] and UV exposure [5]), diet, and lifestyle changes are established contributors to the underlying causation of genomic mutations leading to increased cancer burden and mortality. Over the past decade, advancements in high-throughput molecular profiling by next-generation-sequencing technologies have not only characterized most of the genetic alterations, both driver and passenger, in major types of cancers [6,7,8], but have also identified mutational signatures that are associated with ageing, carcinogenic exposure, or abnormalities in DNA repair and maintenance, which affect genome stability. By accounting for somatic mutations that have accumulated as a consequence of the mentioned processes, large-scale consortia [6,7,8,9] have attributed and revealed the many mutational signatures associated with human cancer types and etiology. This approach also presents mutational signatures with the ability to serve as biomarkers, enabling benefiting opportunities for diagnosis [10] and treatment guidance [11,12,13,14].

Aristolochic acids (AAs) are a family of carcinogenic phytochemical compounds commonly found in plants of the genera Aristolochia and Asarum. Mechanistically, AAs are converted through nitro-reduction to form reactive nitrenium ions of the AA-derived metabolites [15]. The carcinogenicity of AA exposure results from the covalent binding of AA-derived metabolites (aristolactams) [16] with the exocyclic amino groups of purine nucleotides in DNA, resulting in the formation of DNA adducts [17]. The formation and accumulation of these DNA adducts eventually results in the inhibition of effective DNA repair [18]. When coupled with long periods of chronic exposure, the ensuing tumor is usually associated with a unique mutational profile referred to as signature 22 or single-base substitution 22 (SBS22) [9], and harbors an increased mutational load arising from T:A > A:T transversions, particularly in the C[T > A]G trinucleotide context. Comprehensive genomic profiling of hepatocellular cancers (HCC) [19,20], bladder cancers (BCs) [21], renal cell cancers (RCCs) [22], and upper urinary tract urothelial cell carcinoma (UTUC) [23,24,25] has highlighted the widespread prevalence of AA signatures in cancer patients across certain parts of Asia, particularly in Taiwan [20,23]. The latter may be related to high consumption of AA-containing herbal products, highlighted in a 2008 longitudinal study from randomly sampled prescription records of 200,000 patients using data from National Health Insurance in Taiwan. The study showed that at least one-third of the Taiwanese population had been prescribed herbal products containing AA [26].

Exposure to AA is commonly initiated through oral ingestion as it is contained in plants from the genera Aristolochia and Asarum, which are used in traditional and complementary medicines [27,28,29]. After absorption, they are metabolized in the liver and are subsequently excreted through the genitourinary system. Today, evidence has clearly demonstrated the prevalence of AA mutational signatures in cancers of the hepatobiliary [19,20,30] and genitourinary systems [21,22,23,24,25,31]. However, in OGITCs, the prevalence, role, and significance remain unknown. The direct exposure of cells with AA following oral ingestion suggests a possible etiological relationship. In addition, the detection of AA signatures may have potential clinical relevance as these cancers frequently harbor an increased tumor mutation burden (TMB), an established predictive biomarker for checkpoint immunotherapy [19,20].

We therefore performed whole-exome sequencing to systematically investigate for evidence of AA exposure in four different types of OGITCs including oral squamous cell carcinoma (OSCC), esophageal squamous cell carcinoma (ESCC), gastric adenocarcinoma (GA), and colonic adenocarcinoma (CA). We did not uncover significant AA exposure signatures in our cohort, suggesting that AA is not a major carcinogenic factor contributing to the development of OGITCs, in contrast to cancers of the hepatobiliary and genitourinary systems.

## 2. Materials and Methods

### 2.1. Whole Exome Sequencing

Oral squamous cell carcinoma (OSCC), esophageal squamous cell carcinoma (ESCC), gastric adenocarcinoma (GA), and colonic adenocarcinoma (CA) are collectively termed as OGITCs in this study. In this study, fresh frozen tissues were collected from 54 patients from Chang Gung Memorial Hospital at Linkou, Taiwan. The Internal Review Board for Medical Ethics of the Chang Gung Memorial Hospital approved specimen collection procedures (201801743A3C501). The patients were selected at random and histories of intake of AA-containing medicinal products were not available. Genomic DNA isolated from frozen tissue with paired blood was chosen for whole-exome sequencing. The whole-exome sequencing library was constructed using the Human All Exon kit SureSelect Target Enrichment System (Agilent) version 6 prior to sequencing on an Illumina NovaSeq platform with a paired-end configuration of 2 × 150 bp. The sequencing reads were first aligned to the human reference genome NCBI GRC Build 37 using BWA MEM [32], followed by duplicate marking and base-score recalibration using GATK version 4.14 [33] for post-alignment base-recalibration. Somatic mutations were identified using Mutect2 annotated using VEP [34] and vcf2maf [35]. Somatic mutations called with Mutect2 were filtered using the following criteria: identified as PASS from Mutect2, allele frequency of 0.05 and tumor depth of 50 as a cutoff. TMB calculations were estimated based on the number of annotated missense mutations tallied for individual samples before dividing by the number of targets captured by the size of the exome panel. Visualization of annotated variants was performed using CoMut [36]. Copy number variation in each individual sample was inferred using the CNVkit (v.0.9.6) [37] pipeline. Microsatellite status was analyzed using MSIsensor [38], evaluating and scoring microsatellite regions of the genome for instability.

Retrospective datasets referenced in this study was downloaded from the following sequencing repository (i) NCBI SRA BioProject PRJNA758588 for OSCC (ii) EGA-archive EGAS00001002301 for HCC and (iii) NCBI SRA BioProject PRJEB4138 for UTUC.

### 2.2. Neoantigen Prediction

Neoantigen load was analyzed using the NeoPredPipe [39] coupled with ANNOVAR [40] and netMHCpan [41] to predict the presentation of putative neoantigens from filtered mutations called using Mutect2. OptiType was used to predict HLA genotype using sequencing reads derived from the corresponding normal blood samples. We then identified and categorized 9 mer putative neoantigens according to their binding activity derived from netMHCpan results into strong binders (<0.5) and weak binders (≤2). Both strong and weak binders were used as putative neoantigens in the analysis.

### 2.3. Mutational Signatures

Mutational signature analysis was performed using SigProfiler Extractor [9], which extracts de novo mutation signatures based on a matrix generated from a trinucleotide-based context. The tool extracts the optimal number of mutational signatures for each sample, and then decomposes and assigns the mutational signatures extracted de novo into COSMIC signatures.

## 3. Results

### 3.1. Prevalence of AA Mutations in Cancers

AA (Figure 1A) is present in plants from the genera Aristolochia and Asarum used in traditional and complimentary medicine (Figure 1B). AA-DNA adducts have been detected in UTUC [23,24] and RCC [22], and strong mutational signatures of AA exposure have also been discovered in HCC [19,20], BC [21], UTUC [23], and RCC [22] (Figure 1C). A typical mutational profile caused by AA, referred to as signature 22, contains enriched T:A > A:T transversions particularly in the C[T > A]G trinucleotide context with an associated strand bias (Figure 1D). Direct reports of mutational signatures of AA exposure in OGITCs are lacking.

### 3.2. Mutational Signatures in OGITC

A total of 54 OGITC tumor and matched normal samples from Chang Gung Memorial Hospital (CGMH), Taiwan, were included in the study. Genomic profiling was performed using whole-exome sequencing on genomic DNA classified into 15 pairs of CA, 10 pairs of ESCC, 14 pairs of GA, and 15 pairs of OSCC. De novo mutational signatures were extracted using SigProfilerExtractor [9] for all identified single-base substitutions from all 54 samples. Surprisingly, only one sample (GA33) was detected to harbor mutational signatures unique to AA exposure. The case also exhibited mutational signatures SBS1 and SBS5, and, as a result of AA signatures, harbored A-to-T transversions in the C[T > A]G trinucleotide context with a weakly associated strand bias (Figure 2).

Across the cohort of 54 OGITC samples profiled, signatures SBS1 and SBS5, which are representative of clock-like signatures, were common throughout the samples. SBS6 was observed in GA09 and GA23, which is commonly associated with microsatellite instability (MSI). Among the 15 OSCC cases, 9 (60%) showed SBS2 and SBS13 mutational processes caused by the AID/APOBEC family of cytidine deaminases activities. Other signatures detected in this group of samples included SBS10b and SBS18, with a proposed etiology based on reactive oxygen species (ROS) damage, and SBS31, a mutational pattern attributed to exposure to platinum-based chemotherapy.

To investigate mutational signatures associated with the presence of MSI, WES data were used to score MSI status of the tumor with MSIsensor-pro [38]. Samples were considered MSI if unstable microsatellite sites were greater than 20% of the total microsatellite sites. The results obtained from MSIsensor-pro agree with the activity of SBS6 associated with tumors with MSI, a unique signature because of defective DNA mismatch repair (Figure 3A).

We similarly performed mutational signature analysis with an additional dataset of 50 OSCC samples generated previously by Chen et al. [42] (Figure 3B). Consistent with our cohort, no evidence of AA exposure was detected in this dataset, but we observed a mutational signature SBS2 indicative of an APOBEC signature. Our results suggest that AA is probably not the main factor driving tumorigenesis in OGITCs compared to RCC, HCC, BC, and UTUC (Figure 3C).

### 3.3. Somatic Mutation Landscape of OGITC

A total of 7028 missense somatic mutations across 4535 genes were identified across all tumors in this cohort. Across the different OGITCs profiled, the cohort shared similar characteristics to molecular biomarkers observed in TCGA and ICGC cohorts (Figure 4). Recurrent somatic mutations in *TP53* were observed in 27 out of 54 samples (50%) profiled. CA was predominant with mutations in both *TP53* and *APC* with some samples carrying *KRAS,*
*PIK3CA,* as well as *FAT4* mutations. ESCC harbors mutations in the *TP53* gene in 5 out of 10 (50%) samples profiled. GA is primarily characterized by mutations in *TP53, LRP1B,* and frameshift indels in *ARID1A*. In OSCC, the mutations are mainly *TP53,* alongside *FAT1* and *CASP8*.

TMB was evaluated across the cohort by selecting somatic non-synonymous mutations resulting in a protein change. All four different subtypes of OGITC expressed varying levels of TMB, ranging from as low as 0.05 mutations per megabase (Mut/Mb) to as high as 35.7 Mut/Mb. The average TMB profiled across the different OGITC subtypes was 2.59 Mut/Mb for CA, 2.42 Mut/Mb for ESCC, 5.74 Mut/Mb for GA, and 2.21 Mut/Mb for OSCC. GA exhibited the highest TMB, with two samples, GA09 and GA23, having TMB levels of 25.7 and 35.7 Mut/Mb, respectively. These were the only two samples that had TMB greater than 10 Mut/Mb; the samples also presented with microsatellite instability. TMB and MSI results are summarized in Figure 4.

With information derived from HLA genotyping and somatic mutations, neoantigen burden was estimated using NeoPredPipe [39]. Neoantigens, which are mutated peptides, may cause the presentation of tumor-specific peptides capable of initiating an anti-tumor immune response. This is achieved through the prediction of neopeptide binding affinity to the major histocompatibility complex (MHC) region based on 8 to 10 mer epitopes. Analyses were performed over a collection of retrospective cases from previously published literature [20,23,42] related to AA. A swarm plot showing the neoantigen load predicted across different tumors indicates increased presentation of putative neoantigens capable of binding to the MHC region due to increased AA activity (Figure 5).

### 3.4. Copy Number Landscape of OGITC

Variation in copy number was evaluated across the OGITC cohort using CNVkit [37] followed by GISTIC2 [43] to detect significant regions of the genome with aberrant copy number alterations between the different subtypes of OGITC in this cohort. Regions with significant copy number alterations were observed in CA, with amplifications detected in 7p11.2, 18q22.1, and 22q11.21 and deletion in 1q, 3q, 4q, 8p, 9p, 17q, and 22q of the genome. ESCC showed the amplification of a region starting with *CCND1* and ending around *SHANK2* in 11q13.3, and another significant amplification in ESCC was from *PIK3CA* to *SOX2* in 3q26.31, coupled with deletion of *CDKN2A* in 9p21.3. GA showed copy number amplifications in *CDH7/19* in 18q22 and *EGFR* in 7p11, whereas copy number loss was observed in the following regions in the chromosomes 1q, 3q, 4q, 8p, 9p, 17, and 22q. OSCC showed deletion of *CDKN2A/B* in 9p21, and a commonly observed deletion inclusive of the genes from *HSPA7* to *FGCR2C* in 1q23. Deletion in 7q, 8,15q, 17, and 19q was also observed in OSCC (Appendix A).

## 4. Discussion

In this study, we profiled 54 OGITCs from four different sites (oral cavity, esophagus, stomach, and colon) using whole-exome sequencing. De novo mutational signature analysis revealed that only GA33 exhibited AA signatures having characteristics of A-to-T transversions and associated transcriptional strand bias. Amongst the 54 OGITC profiled, we observed only a single case (GA33) suggestive of AA exposure. The observation is interesting and in contrast to that in previous reports (Appendix A) where significant proportions of upper urothelial tract carcinoma (UTUC) [23,44], clear cell renal cell carcinoma (ccRCC) [22], bladder cancer (BC) [21], and hepatocellular carcinoma (HCC) [20] from Taiwan harbor unique AA signatures at significant levels (Appendix A). Therefore, we expected that a proportion of OGITCs from Taiwan would also harbor similar signatures. Additionally, by cross-referencing the OSCC dataset generated previously by Chen et al. [42], no evidence of AA signatures was observed in their study cohort. This implies that DNA adduct damage as a result of mutagenesis from AA exposure probably occurs only at confined well-reported sites of the urothelium, liver, bladder, and kidney. The reasons for this observation remain unclear, though several postulates may be proposed.

The metabolic activation of AA is crucial for the formation of covalent DNA adducts, in which 7-(deoxyadenosin-N6-yl)-aristolactam (dA-AL) has been proven to be a biomarker of AA exposure [45,46,47]. This is supported by in vitro and in vivo findings confirming the increase in AA reactivity following nitro-reduction and sulfation [48]. Nitro-reduction of AA to N-hydroxyaristolactam (AL-NOH) can be catalyzed by different enzymes: CYP1A1, CYP1A2, NAD(P)H: quinine oxidoreductase (NQO1), NADPH P450 reductase (CPR), or prostaglandin H synthase (PHS) [49,50,51]. Apart from CYP1A1/2 and PHS, which are only expressed in liver sites, PHS is prominently expressed in kidney and urothelial tissues [50], whereas NQO1 and CPR are found in other tissues. Most importantly, an ex vivo human liver-kidney model study successfully demonstrated the metabolic activation of AA-I to AL-I-N-OSO_3_H in the liver and its subsequent transport and uptake by the kidney [52], explaining the localization of dA-AL DNA adduct formation at liver, kidney, and urothelial sites.

Two recent publications extensively profiled somatic mutations in normal urothelium [24,31], identifying patterns of AA mutational profiles and suggesting dA-AL DNA adduct formation can occur prior to tumorigenesis as a result of AA exposure. Coupled with chronic AA exposure, periodic accumulation of dA-AL DNA causes high mutagenicity, as whole-genome nucleotide excision repair (NER) is unable to recognize and repair these covalently bonded adducts. The process of cancer initiation by AA exposure is slow and requires constant periodic exposure, as seen in a study of mice fed with AA, where tumors were only detected in the liver about 9 months later [53]. Therefore, whereas cells in OGITCs also harbor enzymes to metabolize AA into dA-AL, the transient exposure and high turnover rate are another explanation for why dA-AL DNA adduct-containing cells are sloughed off prior to a malignant transformation, resulting in the absence of AA signatures in OGITCs. Alternatively, drivers of OGITCs may be selectively advantageous as opposed to mutagenesis induced by the formation of dA-AL DNA adducts upon metabolism of AA, resulting in our current observations.

Apart from etiological relevance, our current findings have therapeutic implications as it is well-established that AA signatures are positively correlated to higher TMB [9,20,31] and, as a result, the presentation of neoantigens derived from tumor somatic mutations. With the use of antigen prediction tools, HCCs with AA signatures were found to have higher presentation of predicted neoantigens as well as infiltrating immune cells [20,54] due to their extremely efficient and persistent mutagenesis as a result of unrepairable dA-AL DNA adduct formation in the genome. Thus, hepatobiliary and genitourinary cancers harboring AA signatures may be more likely to respond to immune checkpoint blockade therapy [55,56,57]. The low TMB found in most OGITCs may be one of the factors that explain the poor responses of patients to immunotherapy strategies.

Our current study is limited by the relatively small sample size, albeit with good representation (10–15 cases of each subtype) of a spectrum of OGITCs. We conclude that AA signatures in OGITC are rare and unlikely to be a major contributing tumorigenic risk factor for OGITC.

## Figures and Tables

**Figure 1 cancers-14-00576-f001:**
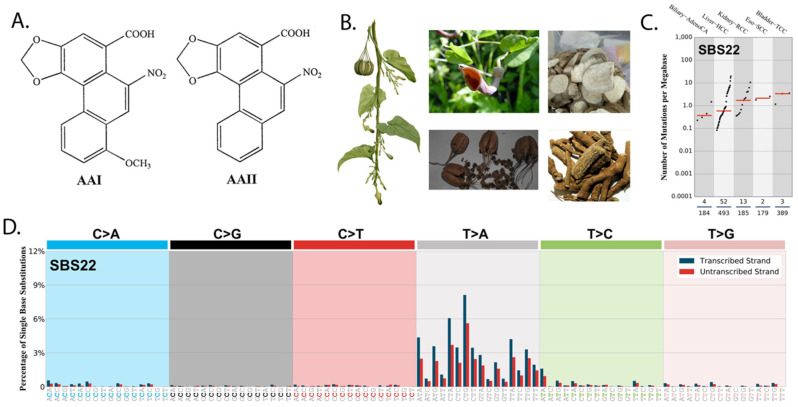
Prevalence of AA mutations in cancers. (**A**) Chemical structure of AA. (**B**) *Aristolochia acuminata* and images of flower, dried fruits, and rhizomes from Aristolochia species containing AA. (**C**) Tumor mutational burden of cancers with presence of AA signature. (**D**) The 384 mutational type classification with an AA signature. (Downloaded from: https://cancer.sanger.ac.uk/cosmic/signatures/SBS/SBS22.tt, accessed on 25 April 2021).

**Figure 2 cancers-14-00576-f002:**
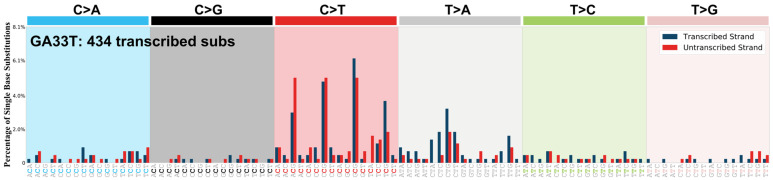
Mutational spectrum for GA33 with A to-T transversions with a weakly associated transcriptional strand bias in the C[T > A]G trinucleotide context.

**Figure 3 cancers-14-00576-f003:**
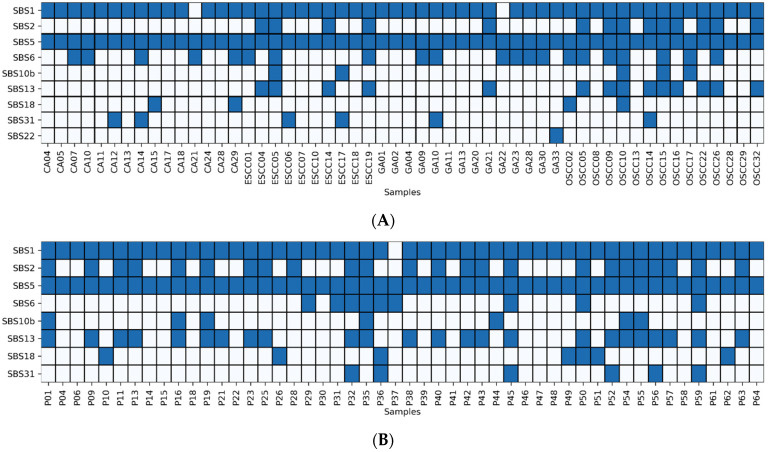
Mutational signatures in OGITCs. (**A**) Heatmap indicating the associated mutational signatures in this study. (**B**) Heatmap of the associated mutational signatures in the OSCC cohort from Chen et al. [42], and (**C**) mutational signatures of 54 OGITCs with proportions inferred from mutational signatures activities grouped to the specific subtype.

**Figure 4 cancers-14-00576-f004:**
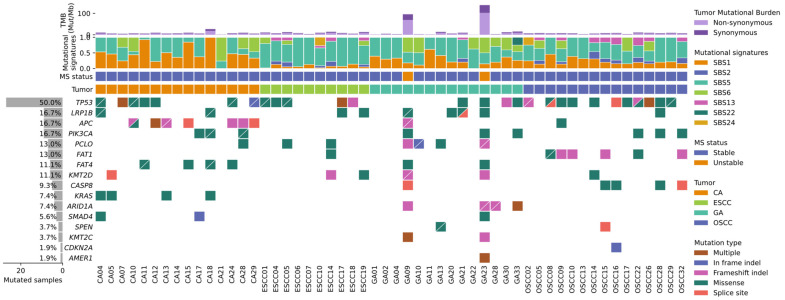
Whole-exome sequence of 54 OGITC with tumor mutational burden (TMB) and MSI status.

**Figure 5 cancers-14-00576-f005:**
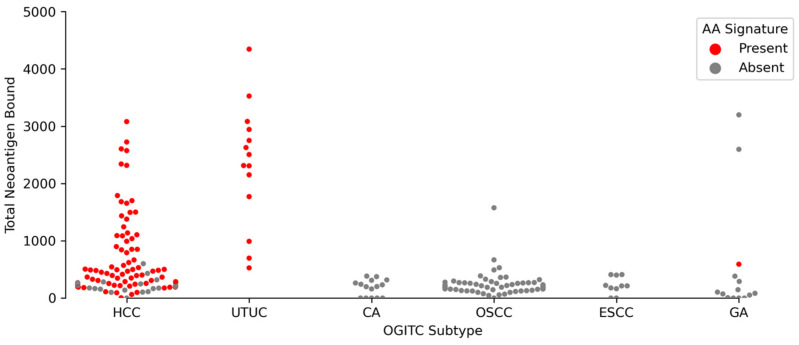
Swarm plot of total neoantigen bound across multiple cancer types. Every point on the swarm plot represents a sample. Red indicates samples with presence of AA signature (SBS22) and grey indicates no presence of AA. Data for HCC [20] and UTUC [23] were obtained from retrospective studies.

## Data Availability

Sequence data for this study were deposited at the European Genome-phenome Archive (EGA), which is hosted by the EBI and the CRG, under accession number EGAS00001005909.

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
