# Peer review of "Rare Occurrence of Aristolochic Acid Mutational Signatures in Oro-Gastrointestinal Tract Cancers"

_cancers, 2022, doi:10.3390/cancers14030576_

Round 1
Reviewer 1 Report
The main aim of the study entitled “Rare occurrence of Aristolochic Acid Mutational Signatures in Oro-Gastrointestinal Tract Cancers” by Lim and colleagues is to investigate the possible association of aristolochic acid-mediated mutagenesis with the development of Oro-Gastrointestinal tract cancers in the Taiwanese population.
Aristolochic acid is a phytochemical compound found in plants of Aristolochia and Asarum genus that are consumed as herbal plant-based remedies by the Taiwanese population. Aristolochic acid is known as a carcinogen as its mutagenesis potential has been demonstrated in animal models and patient tumors particularly hepatocellular carcinoma.
The approach here was to explore the prevalence of aristolochic acid-mediated mutation in tumor mutation burden. To this end, the researchers performed whole-exome sequencing of a total of 54 patients’ tumors from 5 distinct types of oro-gastrointestinal tract cancers and then investigated the incorporation of Single-Base Substitution (SBS) signatures including aristolochic acid mutational signature (SBS22) in the exome of the tumors.
As result, they found high levels of SBS1, SBS5, SBS6, SBS13, and SBS2 signatures highly enriched in the regions corresponding to DNA mismatch repair, reactive oxygen species, and APOBEC family of cytidine deaminases activities. Aristolochic acid mutational signature (SBS22), however, was detected in only one tumor. The researcher also analyzed the mutational signatures of another dataset (Chen et al. 2017) containing whole-exome sequencing data of 50 Oral Squamous Cell Carcinoma (OSCC). None of the tumors from Chen et al. 2017 study showed aristolochic acid mutational signature and the prevalence of other signatures was found highly like that of the current study.
Taken together, the authors conclude that enrichment of aristolochic acid mutational signatures in Oro-Gastrointestinal tract cancers is rare and is unlikely to be a major contributing factor in oro-gastrointestinal tract tumorigenesis.
Overall, this study is a well-designed clinical genetic study and contains important information about the etiology of Oro-Gastrointestinal tract cancers focusing of, this study is well-designed. Hypothesis and rationales are explained clearly by referring to previous findings. The chosen methodology is suitable and results are presented clearly.
However, there are some questions and errors to be addressed:
- Authors are encouraged to add some information about the exposure of the patients to AA if there is any information! Authors should provide evidence that all their patients are exposed to AA. If yes, how often? If no AA exposure, the lack of AA mutational signature is quite normal.
- It would be better to include data of adjacent non-tumor tissues as control. It would help to better identification of tumor-specific mutations and tumor mutation burden.
- Keyword ‘Immunotherapy’ does not fit to this manuscript/study. Please remove it from the list. I recommend adding ‘Aristolochic Acid’ to the list instead.
- In line 36: ‘…exome sequencing to profile 60 cases…’. While the main text talks about 54 tumors.
- The citation is not in format. In the text, reference numbers should be placed in square brackets [ ], and placed before the punctuation; for example [1], [1–3], or [1,3]. On space should be placed between the last world and brackets. Ending dot should be placed after citation not before (please also revise citations in lines 264, 265 and 268).
Author Response
Thank you for giving us the opportunity to improve the manuscript based the valuable comments you have given. We have addressed the comments raised in blue. The below responses have been included in the revised manuscript. In addition to the below comments, formatting guidelines pointed out have been corrected. We look forward to hearing from you to respond to any further questions and comments you may have.
1. Authors are encouraged to add some information about the exposure of the patients to AA if there is any information! Authors should provide evidence that all their patients are exposed to AA. If yes, how often? If no AA exposure, the lack of AA mutational signature is quite normal.
Thank you for the comments, we do acknowledge that the information with regards to patients AA exposure is limited. However, as the design was to investigate the prevalence of AA in cancers from the oro-gastrointestinal tract, the samples in this study were selected at random. Despite the lack of information on this, retrospective study and survey conducted towards the use of AA containing herbal products have highlighted the widespread use resulting in highly prevalent localized cancers of the liver and genitourinary systems. Based on our survey results, we had expected that at least 25% of all the oro-gastrointestinal tract cancers would have evidence of AA exposure. We have included this information in Supp Table 1 (as appended below). Additionally, a study (Hsieh et al, 2008) showed that more than one-third (39.3%) of the population in Taiwan were prescribed with AA between 1997 and 2003. (https://doi.org/10.1186/1749-8546-3-13)
Supp Table 1. Previously reported publications with AA-signatures from Taiwan.
|
Cancer Subtype |
Method |
Cohort Size |
AA positive |
Study |
|
UTUC |
TP53 Amplicon with AL-DNA adducts |
148 |
38 (AL-DNA adduct positive with TP53 A->T transversions) |
Chen at al, 201236 |
|
AA-UTUC |
WES |
9 |
9 |
Poon et al, 201324 |
|
UTUC |
WES |
19 |
17 |
Hoang et al, 201326 |
|
BC |
WGS |
13 |
3 |
Song et al, 201522 |
|
ccRCC |
WES |
10 |
6 |
Hoang et al, 201623 |
|
HCC |
WES |
98 |
76 |
Ng et al, 201721 |
|
UTUC |
WGS |
90 |
27 |
Lu et al, 202025 |
2. It would be better to include data of adjacent non-tumor tissues as control. It would help to better identification of tumor-specific mutations and tumor mutation burden.
During the conceptualization of this study, we did not rule out the use of adjacent non-tumor tissue as a control. However, tissues obtained adjacent from the tumour may not be 100% normal and lie in an intermediary stage between healthy and malignant. Therefore, we believe DNA obtained from peripheral blood is a better control. In our current study, all tumor tissues studied were matched against their paired blood samples, enabling the accurate identification of tumor-specific mutations and tumor mutation burden.
3. Keyword ‘Immunotherapy’ does not fit to this manuscript/study. Please remove it from the list. I recommend adding ‘Aristolochic Acid’ to the list instead.
Thank you for the suggestions, we have added Aristolochic Acid to the keywords for the manuscript.
4. In line 36: ‘…exome sequencing to profile 60 cases…’. While the main text talks about 54 tumors.
Thank you for highlighting the difference in case number between the abstract and the main text. The total case number of oro-gastrointestinal tract cancers profiled in this study was 54.
5. The citation is not in format. In the text, reference numbers should be placed in square brackets [ ], and placed before the punctuation; for example [1], [1–3], or [1,3]. On space should be placed between the last world and brackets. Ending dot should be placed after citation not before (please also revise citations in lines 264, 265 and 268).
We have changed the format of the in-text citation in the revised manuscript.
Reviewer 2 Report
Nice work from Taiwan to examine the role of Aristolochic acid (AA) in Oro-Gastrointestinaltract cancer where the exposure to AA is high because of its use in different alternative medicine. Only a few comments &/or suggestion to improve the paper.
- As the study comments on Aristolochic acid mutation signature, it is essential to elaborate the history of AA use in these patients.
- In the material & Method section: it is not clearly understood how the patients were selected and what was the past history of potential use of AA containing medicinal products. If the selection is random, and if there is no history of AA containing product usage, then the interpretation of this study may be different. But either way, this low prevalence of AA-mutational signature is important.
- Was the DNA adduct measured? If not, why?
- Apart from mentioning only the type of mutational signatures (e.g SBS1, SBS5 etc in Figure 3) in the different pathology (oral, esopgageal, colon etc), it would be better for the readers to get a summary of somatic mutations in different GI cancer in this study. Figure 4 has the information in the total group. Summarizing by oral, esophageal, gastric and colon would be helpful.
Author Response
Thank you for giving us the opportunity to improve the manuscript based on the valuable comments provided. We have addressed the comments below and the responses have been included in the revised manuscript. In addition to the below comments, formatting guidelines pointed out have been corrected. We look forward to hearing from you to respond to any further questions and comments you may have.
1. As the study comments on Aristolochic acid mutation signature, it is essential to elaborate the history of AA use in these patients.
Thank you for the comments, we do acknowledge that information regarding the use of AA containing herbal products in these patients is important to provide a holistic overview. However, the recruitments of samples in this study were selected at random. Retrospective studies have highlighted the widespread use of Aristolochic containing herbal remedies across Taiwan and its mutational signature and abundance in cancers of hepatobiliary and genitourinary systems are highly prevalent. Based on our survey results, we had expected that at least 25% of all the oro-gastrointestinal tract cancers would have evidence of AA exposure. We have included this information in Supp Table 1. Additionally, a study (Hsieh et al, 2008) showed that more than one-third (39.3%) of the population in Taiwan have been prescribed with AA between 1997 and 2003. (https://doi.org/10.1186/1749-8546-3-13)
Supp Table 1. Previously reported publications with AA-signatures from Taiwan.
|
Cancer Subtype |
Method |
Cohort Size |
AA positive |
Study |
|
UTUC |
TP53 Amplicon with AL-DNA adducts |
148 |
38 (AL-DNA adduct positive with TP53 A->T transversions) |
Chen at al, 201236 |
|
AA-UTUC |
WES |
9 |
9 |
Poon et al, 201324 |
|
UTUC |
WES |
19 |
17 |
Hoang et al, 201326 |
|
BC |
WGS |
13 |
3 |
Song et al, 201522 |
|
ccRCC |
WES |
10 |
6 |
Hoang et al, 201623 |
|
HCC |
WES |
98 |
76 |
Ng et al, 201721 |
|
UTUC |
WGS |
90 |
27 |
Lu et al, 202025 |
2. In the material & Method section: it is not clearly understood how the patients were selected and what was the past history of potential use of AA containing medicinal products. If the selection is random, and if there is no history of AA containing product usage, then the interpretation of this study may be different. But either way, this low prevalence of AA-mutational signature is important.
The selection of these patients was random in this study. However, the widespread use of Aristolochic containing herbal remedies are common in prescriptions across Taiwan. Hence, the interpretation of this study in which AA mutational signature is highly likely to be confined to hepatobiliary and genitourinary systems are important in a way which the reactive metabolite Aristolactams are highly likely drive carcinogenesis in those tissue site. However this warrants further investigation.
3. Was the DNA adduct measured? If not, why?
Abundance of DNA adduct was not measured in this study as the objective of this study was to detect the presence of AA mutational Signature (SBS22) in this randomized set of cohort. Evidence establishing the correlation between AA exposure and SBS 22 have also been well replicated in multiple sequencing centres, orthogonal techniques and supporting in-vivo experimental confirmations. (https://cancer.sanger.ac.uk/signatures/sbs/sbs22/)
4. Apart from mentioning only the type of mutational signatures (e.g SBS1, SBS5 etc in Figure 3) in the different pathology (oral, esopgageal, colon etc), it would be better for the readers to get a summary of somatic mutations in different GI cancer in this study. Figure 4 has the information in the total group. Summarizing by oral, esophageal, gastric and colon would be helpful.
Thank you for the recommendation, we have incorporated the contribution of each mutational signature into Figure 4 which will give the reader clarity on the summary of somatic mutation and the contribution from the type of mutational signature.